

# Enhancing wheat production and quality in alkaline soil: a study on the effectiveness of foliar and soil applied zinc

Farhat Ullah Khan[1,*], Adnan Anwar Khan[2,*], Yuanyuan Qu[3], Qi Zhang[3], Muhammad Adnan[4], Shah Fahad[5,6], Fatima Gul[7], Muhammad Ismail[1], Shah Saud[8], Shah Hassan[9] and Xuexuan Xu[1]

[1] Institute of Soil and Water Conservation, Northwest A&F University, Yangling, China
[2] College of Natural Resources and Environment, Northwest A&F University, Yangling, China
[3] College of Grassland Agriculture, Northwest A&F University, Yangling, China
[4] Department of Agriculture, University of Swabi, Swabi, Pakistan
[5] Department of Agronomy, Abdul Wali Khan University, Mardan, Pakistan
[6] Department of Natural Sciences, Lebanese American University, Byblos, Lebanon
[7] Department of Horticulture, University of Agriculture Peshawar, Peshawar, Pakistan
[8] College of Life Science, Linyi University, Linyi, China
[9] Department of Agricultural Extension Education and Communication, The University of Agriculture, Peshawar, Peshawar, Pakistan

[*] These authors contributed equally to this work.

Corresponding authors
Shah Fahad,
shah_fahad80@yahoo.com
Xuexuan Xu, xuxuexuan@nwsuaf.edu.cn

## ABSTRACT

Cultivation of high-yield varieties and unbalanced fertilization have induced micronutrient deficiency in soils worldwide. Zinc (Zn) is an essential nutrient for plant growth and its deficiency is most common in alkaline and calcareous soils. Therefore, this study aimed to evaluate the effect of Zn applied either alone or in combination with foliar application on the quality and production of wheat grown in alkaline soils. Zn was applied in the form of zinc sulfate ($ZnSo_4$) to the soil and as a foliar spray during the sowing and tillering stages, respectively. Results showed that Zn fertilization of wheat, irrespective of modes of application, significantly increased grain and biological yield, grain per spike, and 1,000 grains weight over control; however, its effect was more noticeable when applied as 7.5 kg ha$^{-1}$ of soil Zn combined with foliar Zn at 2.5 kg ha$^{-1}$. Zn application significantly increased the grain protein content from 9.40% in the control to a maximum of 11.83% at soil Zn of 10 kg ha$^{-1}$. Similarly, Zn application improved Zn, phosphorus (P), and potassium (K) concentrations in wheat grains. Moreover, correlation analysis showed that the grain Zn concentration was positively correlated with the grain P concentration. The correlation between P concentration in wheat grains and 1,000 grain weight was not significant. A total of 1,000 grains weight was positively correlated with tillers per plant, grain yield, and biological yield. There were positive correlations between protein content, biological yield, grain yield, and tillers per plant. Therefore, soil-applied Zn + foliar application in alkaline soils with limited Zn availability is crucial for improving wheat yield and grain quality.

## INTRODUCTION

Wheat (*Triticum aestivum* L.) belongs to the family Poaceae and is a key cereal crop for the vast majority of people worldwide. For almost 36% of the global population (two billion people), wheat is the most favored staple food. Wheat accounts for nearly 55% of all carbohydrates and 20% of all calories consumed worldwide (*Azeem et al., 2019*; *Jat et al., 2018*). Pakistan is one of the top wheat-producing countries in the world, ranking 8th and accounting for approximately 3.17% of the total global wheat production (*Sher et al., 2022*). The substantial production contribution of wheat makes it a crucial food grain crop for the country's economy. The grain yield of wheat is affected by several biotic and abiotic factors. Among abiotic factors, one of the primary causes of low yield is nutritional imbalance and lack of essential macro-and micronutrients (*Azeem et al., 2019*). To some extent, wheat yield has improved owing to the application of mineral nitrogen (N) and phosphorus (P), but the potential yield has not been achieved in Pakistan (*Khan et al., 2017*). By 2050, the world's population is expected to reach 9.8 billion, presenting a major challenge for scientists to not only increase food production, but also improve the nutritional quality of the food produced, especially in developing regions (*Ahmad et al., 2023*).

Alkaline calcareous soils in developing countries, owing to poor management, have resulted in reduced crop productivity and quality (*Bhatt, Hossain & Sharma, 2020*). Zinc deficiency is most common in alkaline and calcareous soils. Most soils in Pakistan are calcareous, which can cause some nutrients to become unavailable owing to factors such as high pH, low organic matter, salt stress, and imbalanced application of mineral fertilizers (*Hafeez, Khanif & Saleem, 2013*). These factors collectively contribute to Zn deficiency in soils. It is now an established fact that Pakistan's soil is deficient in zinc, which can cause a reduction in the yield of certain cereal crops, such as wheat. Furthermore, those who consume a diet high in cereals may experience health issues linked to micronutrient deficiencies (zinc) such as impaired brain and immune function, stunted growth in children, harm to physical development, weakened resistance to disease, unfavorable pregnancy outcomes in women, and increased morbidity and mortality (*Cakmak, 2008*). In Pakistan, approximately 12 million children suffer from stunted growth, whereas zinc deficiency affects 22.1% of women and 18.6% of children under the age of five (*Sher et al., 2022*).

Zn was the first micronutrient recognized as deficient element in Pakistani soils, and later studies confirmed the widespread zinc deficit in all rice-growing regions (*Yoshida & Tanaka, 1969*). A study conducted in alkaline soils of Pakistan reported that Zn applied at 5 kg ha$^{-1}$ and 15 kg ha$^{-1}$ increased wheat yield by 18% and 41% over control, respectively (*Khattak, Dominy & Ahmad, 2015*). They also observed that soil Zn application combined with foliar spraying increased the wheat grain protein content by 29.5%. Reducing Zn deficiency in plants can be achieved through a variety of practices, such as the supplementation of mineral Zn fertilizer through soil and foliar spray (*Peter et*

*al., 2017*). Zinc foliar application is a technique used to enhance the zinc status of plants by externally applying solid or liquid zinc fertilizers to crop leaves at the appropriate growth stage. Another previously published study demonstrated that combined application of soil Zn and foliar spray did not significantly affect grain yield and biomass of wheat crop in alkaline soil of loess plateau of China; however, Zn foliar spray significantly increased Zn grain concentration by 28% and 89% during first and second season of wheat (*Wang et al., 2012*). Conventionally, mineral Zn fertilizers are the most commonly used fertilizer because of their low cost and high solubility (*Roohani et al., 2013*). However, fixation reactions can decrease the bioavailability of zinc in soils due to high carbonate content and high pH. This can lead to the adsorption on calcite or the precipitation of $Zn(OH)_2$ or $ZnCO_3$, making conventional fertilizers ineffective for crop zinc uptake (*Recena, García-López & Delgado, 2021*; *Rehman et al., 2020*). In alkaline soils pH condition, Zn has a strong affinity for clay adsorption which tends to accumulate unusable Zn hydroxides on the surface of clay minerals (*Walaszek et al., 2018*).

The size of the accessible Zn pools in the soils had a significant impact on the Zn concentration and yields of the plants. The majority of cereal-growing regions face chemical and physical obstacles that considerably decrease the access of Zn to the plant roots. Thus, strategies are required to enhance Zn availability in soils and to improve cereal grain quality and production. Many researchers have studied the effect of Zn fertilizer on crop yield; however, limited information is available on soil Zn fertilizer application combined with foliar spraying in alkaline soils in Pakistan. Considering the above facts, the current study aimed to address the limited availability of Zn in alkaline soils. This was achieved by evaluating the following set of objectives: (i) to evaluate the effect of Zn application to soil, either alone or in combination with foliar spraying, on the quality and yield of wheat grown in alkaline soil; (ii) to determine the optimal rate of Zn application for maximum wheat yield and quality in alkaline soil; and (iii) to investigate the effect of Zn application on the protein content of wheat grains in alkaline soils.

# MATERIALS AND METHODS

## Experimental site
The experimental site is located in Kohat region (33°30'22.4"N 71°26'06.0"E) of Khyber Pakhtunkhwa, Pakistan. The study site has a semi-arid and subtropical climate, with an average annual rainfall of 500–600 mm. Meteorological data were obtained from the Pakistan Meteorological Department website (https://www.pmd.gov.pk/en/). During winter (November–May), the mean maximum and minimum temperatures were 20 and 39 °C, respectively. Physiochemical properties of the experimental site were measured according to the standard protocols, which has been mentioned in section 2.4. Soil pH was 8.1 and EC was 0.32 $dSm^{-1}$, which showed that the soil was alkaline in nature. Soil texture was clay loam with 22%, 38%, and 40% silt, sand and clay, respectively. Soil organic matter, N, P and K content were 0.69%, 0.8 $g\ kg^{-1}$, 1.5 $mg\ kg^{-1}$, 55 $mg\ kg^{-1}$, respectively. The soil AB-DTPA extractable Zn content was 0.58 $mg\ kg^{-1}$, which showed that the soil was Zn deficient.

## Experimental procedure

A field experiment was carried out to assess the impact of soil and foliar applied zinc on the growth, yield, and quality of wheat crop during winter season 2020–2021. A Randomized Complete Block Design (RCBD) was used with three replications. The area of each plot was 9 m$^2$ (3 m ×3 m), and there were total of 33 plots. Wheat was sown at 120 kg ha$^{-1}$ on November 22, 2020, and harvested on May 5, 2021. There were 11 treatments including; control (T1), soil Zn 5 kg ha$^{-1}$ + 0 kg ha$^{-1}$ foliar Zn (T2), soil Zn 3.75 + 1.25 foliar Zn (T3), soil Zn 2.5 + foliar Zn 2.5 (T4), soil Zn 1.25 + foliar Zn3.75 (T5), soil Zn 0 + foliar Zn 5 (T6), soil Zn 10 + foliar Zn 0 (T7), soil Zn 7.25 + foliar Zn2.50 (T8), soil Zn 5 + foliar Zn 5 (T9), soil Zn 2.5 + foliar Zn 7.5 (T10) and soil Zn 0 + foliar Zn 10 (T 11). Zn was applied in the form of zinc sulfate (ZnSO$_4$) to the soil and as a foliar spray during the sowing and tillering stages, respectively. Each treatment received recommended dose of N-P-K at 120:90:60, in the form of urea, diammonium phosphate (DAP), and sulfate of potash (SOP), respectively, at the time of wheat sowing. All agronomic practices were timely done during wheat growth period. To measure wheat agronomic attributes at harvesting stage, five plants were randomly selected from each treatment and average of tillers per plant was calculated. 1,000 grains weight were randomly collected from each treatment using a grain counter machine and weighed using a weight balance. Grain yield was obtained by harvesting the four central rows of each plot, threshed using a mini wheat thresher, dried, and then weighed. Biological yield obtained by harvesting mature wheat from four central rows of each plot, then dried and weighed. Grain yield and biological yield were converted into kg ha$^{-1}$ using the following formulas:

$$Biological\ yield\ (mg\ kg^{-1}) = \frac{Yield\ in\ selected\ rows\ \times\ 10000}{R\ to\ R\ distance\ \times\ row\ length\ \times\ no\ of\ rows\ selected}$$

$$Grain\ yield\ (mg\ kg^{-1}) = \frac{Yield\ in\ selected\ rows\ \times\ 10000}{R\ to\ R\ distance\ \times\ row\ length\ \times\ no\ of\ rows\ selected}.$$

## Soil and plant samples analysis and calculations

Soil samples were collected from 0–20 cm depth with the help of Auger from three different locations in each plot before wheat sowing. Visible plant residues and roots were removed by hand and stored in labeled plastic bags for laboratory analysis. Soil samples were air-dried and passed through a 2-mm sieve. Similarly, plant samples were collected randomly from each plot at wheat harvesting stage. The plant samples were cut into small pieces, air-dried, and stored for further analysis. Soil standard procedures were used to determine the soil pH (*McLean, 1982*) and electrical conductivity (EC) (*Rhoades, 1996*). The methods of *Bouyoucos (1962)* and *Nelson & Sommers (1996)* were used to determine the texture and organic matter content, respectively. The method suggested by *Soltanpour & Schwab (1977)* was used to measure AB-DTPA extractable P, K, and Zn content in the soil (before wheat sowing). Briefly, AB-DTPA extractable Zn was determined by adding 10 g air-dried soil and 20 ml AB-DPTA extracting solution. The mixture was shaken for 15 min and filtered through a 42-wattman filter paper. The supernatant was used to measure P, K, and Zn content is soil samples using a UV–VIS spectrophotometer set at 880 nm, flame

photometer, and atomic absorption spectrophotometer, respectively. The concentration of each element was calculated in soil samples by the following formula:

$$P, K \ and \ Zn \ (mg \ kg^{-1}) = \frac{Instrumental \ reading \ \times \ Volume \ made}{Soil \ sample \ weight \ (g)}.$$

In wheat grain, the Zn concentration was determined using the procedure of *Soltanpour & Schwab (1977)*. Briefly, 1 g dried wheat grain sample was taken in a crucible and ashed at 550 °C. After ashing, it was cooled to room temperature. 10 ml of 0.5 N HCl was then added and filtered through 42-wattman filter paper. An atomic absorption spectrophotometer was used to measure the Zn concentration in the plant samples. Zn concentration was calculated in extract as:

$$Zn \ (mg \ kg^{-1}) = \frac{Instrumental \ reading \ \times \ Volume \ made}{Grain \ sample \ weight \ (g)}.$$

The total P and K concentration in wheat grain were determined using the method described by *Soltanpour & Schwab (1977)*. Briefly, a dried wheat grain sample (0.5 g) was placed in a conical flask, and 10 ml concentrated $HNO_3$ was added. After 24 h, 4 ml perchloric acid was added and heated on a hot plate until white fumes appeared. After cooling, the sample was diluted by adding 100 ml distilled water and filter through 42-wattman filtered paper in a 100 ml volumetric flask. For P determination, 1 ml filtrate in a 25 ml volumetric flask was taken and 5 ml ascorbic acid mix reagent (ammonium molybdate + antimony potassium tartrate) was added. The volume was adjusted to 25 ml and the sample was kept in the dark to develop a blue color. A visible UV–VIS spectrophotometer set at 880 nm was used to measure the P concentration in wheat grain. A flame photometer was used to measure the K concentration in wheat grain. P and K content were determined by the following formulas:

$$P \ (mg \ kg^{-1}) = \frac{Instrumental \ reading \times Volume \ made}{Grain \ sample \ weight \ (g) \ \times \ Volume \ taken}$$

$$K \ (mg \ kg^{-1}) = \frac{Instrumental \ reading \ \times \ Volume \ made}{Grain \ sample \ weight \ (g)}.$$

The protein content in the wheat grain was measured according to *Khattak, Dominy & Ahmad (2015)*. Sodium dodecyl sulfate-polyacrylamide gel electrophoresis (SDS-PAGE) was used to study the effect of zinc on the wheat protein content. Grain flour samples (0.5 g) were obtained in triplicate to determine the grain protein content. After adding 0.25 mL of protein buffer, the sample was mixed with vortex mixture, then the mixture was centrifuged for 15 min at 13,000 rpm. Finally, the supernatant was measured at E-280 nm in mg/mL, or nearly 1 optical density (OD) using UV–VIS spectrophotometer. The protein percentage was calculated by multiplying the absorbance with dilution factor.

$$\% \ protein = absorbance \ reading \times dilution \times \frac{100}{1000}.$$

## Statistical analysis

All collected data were statistically analyzed by one-way ANOVA and means were compared using the Duncan test ($P < 0.05$) in SPSS (Ver; 20.0). Pearson's correlations analysis was conducted using Microsoft excel 2016.

**Table 1** Wheat grain, biological yield, tillers per plant, and thousand grain weight as affected by different levels of soil and foliar applied Zn.

| Zn (kg ha$^{-1}$) | | Grain yield (kg ha$^{-1}$) | Biological yield (kg ha$^{-1}$) | Tillers per plant | 1000 grains weight (g) |
|---|---|---|---|---|---|
| Soil | Foliar | | | | |
| 0 | 0 | 2712.67 f | 6033.33 g | 6.59 d | 40.36 c |
| 5 | 0 | 2928 e | 6422.33 f | 7.22 abc | 43.57 bc |
| 3.75 | 1.25 | 3130 d | 6921.67 e | 7.07 abc | 45.11ab |
| 2.5 | 2.5 | 3237 d | 7034.67 e | 6.69 d | 46.08 ab |
| 1.25 | 3.75 | 3252 d | 7246.67 d | 6.87 cd | 46.68 ab |
| 0 | 5 | 3378 c | 7387.67 d | 6.89 cd | 47.38 ab |
| 10 | 0 | 3454.67 c | 7631 c | 7.02 bcd | 47.47 ab |
| 7.5 | 2.5 | 3639.67 a | 8002.33 a | 7.44 ab | 48.37 ab |
| 5 | 5 | 3608.67 ab | 7994 a | 7.50 ab | 47.15 ab |
| 2.5 | 7.5 | 3588 ab | 7890.33 ab | 7.44 ab | 45.38 ab |
| 0 | 10 | 3504 bc | 7786.33 bc | 7.56 a | 47.0 ab |

Notes.
*Values carrying different letter (s) in columns are significantly different at $P \leq 0.05$.

# RESULTS

## Agronomic attributes

The results indicate that certain rates of zinc fertilizer application, whether soil applied or foliar, had a positive impact on both biological yield and grain yield of wheat (Table 1). Regardless of the method used, Zn application had a considerable positive impact on wheat grain yield. The plot with the maximum wheat grain yield (3,639 kg ha$^{-1}$) received soil + foliar Zn at rates of 7.5 kg ha$^{-1}$ and 2.5 kg ha$^{-1}$, respectively, followed by plot that received soil Zn at 5 kg ha$^{-1}$ + 5 kg ha$^{-1}$ foliar Zn with an average yield (3,608 kg ha$^{-1}$). The lowest grain yield (2,712 kg ha$^{-1}$) was observed in the control. Similarly, the highest biological wheat yield (8,802 kg ha$^{-1}$) was found in the plot where Zn was applied as soil and foliar at 7.5 kg ha$^{-1}$ and 2.5 kg ha$^{-1}$, respectively, followed by plot that received 5 kg ha$^{-1}$ of Zn fertilizer in both soil and foliar, with an average biological wheat yield of 7,994 kg ha$^{-1}$, whereas the lowest biological wheat yield was observed in control plot. The outcome of our experiment predicted that the number of tillers per plant in $T_{11}$ (foliar Zn at 10 kg ha$^{-1}$) was significantly different compared to the control ($P < 0.05$) as shown in Table 1. The maximum number of tillers per plant ($n = 7.56$) was found in $T_{11}$, followed by the plot that were administered zinc through soil and foliar (5:5 kg ha$^{-1}$) producing ($n = 7.5$) tillers per plant. The number of tillers per plant were higher in plots where the foliar ratio of Zn was higher than soil. The results showed that Zn addition increased the 1,000 grains weight but did not show a significant difference among treatments, except $T_1$ (control) which showed the lowest 1,000 grains weight. Data on 1,000 grains weight ranged from 40.36 to 48.37 g. The maximum 1,000 grains weight was recorded in $T_8$ (48.37 g) and $T_7$ (47.47 g), followed by $T_6$ (47.38 g), whereas the minimum (40.35 g) was recorded in $T_1$.

**Table 2   Grain Zn, K and P as affected by different levels of Zn as soil and foliar.**

| Zn (kg ha$^{-1}$) | | | Zn | K | P |
|---|---|---|---|---|---|
| Soil | Foliar | | | Concentration (%) | |
| 0 | 0 | | 0.31 d | 0.50 cd | 0.12 c |
| 5 | 0 | | 0.44 b | 0.51 a | 0.13 bc |
| 3.75 | 1.25 | | 0.45 b | 0.49 cd | 0.13 bc |
| 2.5 | 2.5 | | 0.44 b | 0.48 d | 0.14 ab |
| 1.25 | 3.75 | | 0.46 b | 0.47 d | 0.14 ab |
| 0 | 5 | | 0.44 b | 0.52 a | 0.14 ab. |
| 10 | 0 | | 0.45 b | 0.45 b | 0.14 ab |
| 7.5 | 2.5 | | 0.49 a | 0.50 cd | 0.14 ab |
| 5 | 5 | | 0.39 c | 0.50 cd | 0.15 a |
| 2.5 | 7.5 | | 0.39 c | 0.49 cd | 0.15 a |
| 0 | 10 | | 0.39 c | 0.50 cd | 0.15 a |

Notes.
*Values carrying different letter (s) in columns are significantly different at $P \leq 0.05$.

## Grain Zn, P and K concentrations

The findings regarding wheat grain Zn, K, and P concentrations affected by Zn addition at various levels through the soil and foliar are presented in Table 2. The significant higher Zn concentration (0.49%) was recorded in soil-applied Zn + foliar spray (soil 7.5 kg ha$^{-1}$+ foliar 2.5 kg ha$^{-1}$), followed by 0.46% in soil Zn + foliar application (1.25 kg ha$^{-1}$ + 3.75 kg ha$^{-1}$). The minimum Zn concentration in wheat grains was recorded in control. The effect of Zn supplementation either through soil or foliar application was non-significant on P grain concentration; however, plots that received Zn supplementation had the highest P concentration compared to the control. Zn fertilizer levels had a considerable impact on the potassium concentration of wheat grains. The potassium concentration in the wheat grain increased by 6.24% with foliar Zn at 5 kg ha$^{-1}$($T_6$) compared to control. $T_6$ was significantly different from all the treatments except $T_2$ (soil-applied Zn at 5 kg ha$^{(-1)}$. The plot that received Zn only through the soil at a rate of 10 kg ha$^{-1}$ had the lowest wheat grain K concentration.

## Grain protein content

Zinc fertilization, whether applied through foliar or soil, significantly increased the protein content of wheat grains (Table 3). There were significant differences among the means of some treatments, whereas others showed a slightly increase than the control. $T_7$ (soil Zn 10 kg ha$^{-1}$) treatment showed a 25.88% increase over the control, followed by T8 (soil 7.50 kg ha$^{-1}$ + foliar 2.50 kg ha$^{-1}$), which increased grain protein content by 22%. T2-T10 showed an increase in protein content from 2.48% to 25.88% compared with control. The data showed that increasing soil Zn with decreasing foliar levels gradually enhanced the protein content in the grain.

## Pearson's correlation analysis

Correlation analysis showed that grain Zn concentration was positively correlated with P concentration in the wheat grains (Table 4). The correlation between Zn concentration in

**Table 3  Protein in grain as affected by different levels of Zn as soil and foliar application.**

| Zn (kg ha⁻¹) | | Protein in grain | Increase over control |
|---|---|---|---|
| Soil | Foliar | % | (%) |
| 0 | 0 | 9.40 de | |
| 5 | 0 | 9.63 de | 2.48 |
| 3.75 | 1.25 | 9.83 de | 4.6 |
| 2.5 | 2.5 | 10.13 bcd | 5.32 |
| 1.25 | 3.75 | 10.20 bcd | 8.51 |
| 0 | 5 | 10.53 bcd | 12.05 |
| 10 | 0 | 11.83 a | 25.88 |
| 7.5 | 2.5 | 11.46 ab | 21.99 |
| 5 | 5 | 11.23 abc | 19.5 |
| 2.5 | 7.5 | 10.96 bcd | 16.67 |
| 0 | 10 | 10.76 bcd | 14.54 |

Notes.
*Values carrying different letter (s) in columns are significantly different at $P \leq 0.05$.

**Table 4  Pearson's correlation coefficients.**

| | 1000 grains weight | Tillers per plant | Grain yield | Biological yield | Grain protein | Grain K concentration | Grain P concentration | Grain Zn concentration |
|---|---|---|---|---|---|---|---|---|
| 1000 grains weight | 1 | 0.349* | 0.649** | 0.645** | 0.459** | −0.008 | 0.333 | −0.474** |
| Tillers Per Plant | | 1 | 0.604** | 0.597** | 0.324 | 0.247 | 0.438* | −0.608** |
| Grain yield | | | 1 | 0.971** | 0.631** | 0.039 | 0.678** | −0.832** |
| Biological yield | | | | 1 | 0.652** | 0.068 | 0.739** | −0.838** |
| Grain protein | | | | | 1 | 0.348* | 0.472** | −0.466** |
| Grain K concentration | | | | | | 1 | −0.21 | 0.048 |
| Grain P concentration | | | | | | | 1 | −0.685** |
| Grain Zn concentration | | | | | | | | 1 |

Notes.
**indicate significance at 95% level. Zn, K, and P are concentration in wheat grain.

wheat grains and K was not significant. The wheat grain K concentration was negatively correlated with the grain P concentration. The correlation between P concentration in wheat grains and 1,000 grains weight was not significant. The correlation between the grain K concentration and wheat grain yield was not significant. 1,000 grain weight was positively correlated with number of tillers per plant, grain yield, and biological yield. There were significantly positive correlations between protein content, biological yield, grain yield, and 1,000 grain weight.

## DISCUSSION

### Wheat yield and yield components

Zn was applied through soil and foliar at 7.5 kg ha$^{-1}$ and 2.5 kg ha$^{-1}$, respectively, which greatly improved wheat grain production. These results highlight the significance of zinc availability in increasing wheat yield in alkaline soils. A published study reported that in alkaline soils of Pakistan, adding Zn application in soil at 5 kg ha$^{-1}$ or two split doses as a foliar spray increased maize grain yield by 29% and 14%, respectively (*Khattak et al., 2006*). The results of the current study are similar to those of other studies, which showed that the combined application of Zn through soil + foliar application was more effective in increasing grain production and plant growth than Zn soil application alone (*Imran & Rehim, 2017*). With soil + foliar treated plots, better photosynthetic activity and biomass accumulation resulted in better yields, which could be explained by a sufficient Zn supply and an increased in several soil enzymatic activities, which in turn led to greater wheat yields (*Hussain & Yasin, 2004*; *Paramesh et al., 2014*). Additionally, the Zn concentration in plants increases photosynthesis, carbohydrate transformation, and seed development (*Alloway, 2008*). Other authors have also shown the benefits of using Zn fertilization in improving wheat growth and yield components (*Cakmak, Pfeiffer & McClafferty, 2010*; *Nawaz et al., 2015*). A significantly greater biological yield was observed when foliar Zn fertilizer was supplied along with soil Zn application, which might be due to the response of the wheat crop to foliar Zn application. The fact that zinc is known to be crucial for different plant enzymes as a metal component or as a functional, structural, or regulatory co-factor which may support the increase in biological output caused by zinc nutrition (*Singh et al., 2018*). Another study also reported that Zinc is an essential ingredient in the production and breakdown of proteins, nucleic acids, and carbohydrates (*Cakmak, 2008*). Thus, Zn plays a crucial role in biomass synthesis as a co-factor of auxins and lipids (*Chattha et al., 2017*; *Hassan et al., 2019*). Furthermore, soil Zn with foliar treatment showed greater plant chlorophyll content and biological outcome production in maize by 75 and 54%, respectively (*Mosanna & Behrozyar, 2015*). According to a previous study, maintaining protein synthesis in wheat meristems demands a high Zn concentration (*Mat Hassan et al., 2012*). Our results are supported by prior research who recorded highest wheat straw yield (6.92 t ha$^{-1}$) with both soil Zn + foliar application (*Paramesh et al., 2020*). The number of tillers per plant were more in plots where the foliar ratio of Zn was kept higher than that of the soil. Our findings are in accordance with previous study results who reported that foliar zinc application improved the number of tillers per plant (*Soleimani, 2006*). Zn fertilizer application increased the proportion of productive tillers, which could be due to the availability of additional nutrients to the growing tillers (*Gul et al., 2011*; *Ramzan et al., 2020*). Our results showed that the different treatments of Zn did not significantly increase 1,000 grain weight, except soil Zn7.5 + foliar Zn2.5 treatment. Zinc is required for growth hormone synthesis, starch production, and maturation, all of these plant factors contribute to seed weight gain (*Sarkar, Mandal & Kundu, 2007*). Zinc fertilization increased 1,000-grain weight, possibly because of the strong mobility of zinc within the plant from leaves to stems, roots, and grains (*Rengel, 2001*). Another study also found that

1,000 grains weight was increased substantially through Zn fertilizer application (*Torun et al., 2001*).

## Plant nutrient concentration in wheat grain

Data regarding grain Zn, K, and P as affected by Zn addition at various levels through soil and foliar are shown in Table 2. The Zn concentration in wheat grain was significantly increased by combining soil-applied Zn with foliar spray (soil 7.25 kg ha$^{-1}$ + foliar 2.5 kg ha$^{-1}$). Other investigators have also noted an increase in Zn concentration in wheat grains by Zn fertilization (*Ghasal et al., 2017*; *Niyigaba et al., 2019*). The fact that Zn is mobile in plant which can easily transfer from vegetative portions into emerging wheat grains, may be one explanation for the increase in wheat grain Zn concentration due to foliar spray (*Esfandiari et al., 2016*; *Li et al., 2014*). Furthermore, we found that wheat grain P concentration significantly increased with Zn fertilizer either through soil or foliar application compared to control. A previous study conducted in alkaline soils reported that P concentration in maize leaves and grains increased with increasing zinc external supplementation, however, after attaining the maximum value of P with soil Zn 10 kg ha$^{-1}$ + one foliar spray resulted a higher P concentration in leaves, while soil Zn application combined with two foliar sprays application showed a decrease in wheat grain P concentration by −13.5% (*Khattak et al., 2006*). The possible reason may be that zinc is friendly with P up to the wheat booting stage and translocate parallel to each other, but after depletion in plants, they hinder each other translocation to the given part if one increases the other decreases, and *vice versa*. Our results are in line with other studies, which observed that soil-applied Zn combined with foliar spray had the highest P concentration in wheat grains than soil-applied Zn alone (*Paramesh et al., 2014*). T$_6$ (foliar Zn at 5 kg ha$^{-1}$) in the current study had a significant effect on the K concentration of wheat grains compared to the other treatments, except T$_2$ (soil-applied Zn at 5 kg ha$^{-1}$). The increase in K content of wheat grains with soil Zn + foliar spray might be due to the zinc synergistic relationship with potassium (*Razzaq et al., 2013*). Zn application through foliar spraying has been shown to improve the macronutrient components of plants (*Sayed, Solaiman & Abo-El Komsan, 2004*). The results of another study also reported that Zn application increased maize grain Mg and K concentrations (*Szakal, 1989*).

## Grain protein content

Plant grain protein content is an important quality parameter of wheat. In the current study, wheat grain protein content was increased as a result of the combined application of soil and foliar spray compared to control. The findings suggest that Zn is one of the key nutrients for improving the protein content of wheat grains. Zinc is a stimulating component that enhances indoleacetic acid synthesis by increasing protein and amino acid content of grains (*Paramesh et al., 2020*). A previous study conducted in alkaline soils found that Zn fertilizer application increased wheat grain protein content, with the most effective treatment being Zn at 5 kg ha$^{-1}$ applied to the soil with one foliar spray (*Khattak, Dominy & Ahmad, 2015*). Our findings are consistent with those previously reported studies, which showed that soil Zn application at 25 kg ha$^{-1}$ resulted in an increase (11.8%) in wheat

grain protein content compared to the control (*Paramesh et al., 2014*). The presence of Zn in plant can change the protein composition of wheat grains (*Peck, McDonald & Graham, 2008*). Zn also increase wheat β-Carotene content, which improves grains protein and other quality characteristics (*Kharub & Gupta, 2003*). Another researcher documented that during the seed formation, maximum zinc uptake and protein synthesis happen at the same time (*Ozturk et al., 2006*). Soil Zn application with foliar spray increased dry matter building and N uptake in wheat crop, which in turn raised the protein content of grains (*Barunawati et al., 2013*). Zn shortage is linked to N metabolism; when Zn absorption in plants is reduced, the amount of protein in the plant reduces dramatically (*Gupta, Ram & Kumar, 2016*). Thus, protein synthesis requires appropriate Zn supplementation (*Patel, Kumar & Durani, 2007*).

## CONCLUSIONS

The application of Zn, whether through foliar spray or soil application was found to benefit both the yield and quality of wheat grown in alkaline soils. The highest grain yield, biological yield and 1,000 grain weight were found when soil Zn was applied at 7.5 kg ha$^{-1}$ with foliar at 2.5 kg ha$^{-1}$. The wheat grain Zn concentration was significantly increased over control when soil Zn was combined with foliar spray (7.5 kg ha$^{-1}$ + 2.5 kg ha$^{-1}$). Wheat grain potassium and phosphorus concentrations were enhanced by Zn addition to the crop regardless of the application method. The Zn application increased the wheat grain protein content from 9.40% in control to 11.83% at soil Zn of 10 kg ha$^{-1}$. Thus, soil-applied Zn with foliar spray is recommended to increase wheat grain production and quality in alkaline soils. Further research could be conducted to optimize the timing, dosage, and method of zinc application to crops in order to maximize its efficiency and minimize any potential negative effects. Additionally, the use of Zn in agricultural practices could be expanded to increase the productivity and quality of crops other than wheat, particularly in regions with Zn-deficient soils.

### Funding
The authors received no funding for this work.

### Competing Interests
The authors declare that there are no competing interests.

### Author Contributions
- Farhat Ullah Khan conceived and designed the experiments, performed the experiments, analyzed the data, authored or reviewed drafts of the article, and approved the final draft.
- Adnan Anwar Khan conceived and designed the experiments, performed the experiments, analyzed the data, authored or reviewed drafts of the article, and approved the final draft.

- Yuanyuan Qu conceived and designed the experiments, performed the experiments, analyzed the data, authored or reviewed drafts of the article, and approved the final draft.
- Qi Zhang conceived and designed the experiments, performed the experiments, analyzed the data, authored or reviewed drafts of the article, and approved the final draft.
- Muhammad Adnan conceived and designed the experiments, performed the experiments, prepared figures and/or tables, and approved the final draft.
- Shah Fahad conceived and designed the experiments, performed the experiments, authored or reviewed drafts of the article, and approved the final draft.
- Fatima Gul performed the experiments, prepared figures and/or tables, and approved the final draft.
- Muhammad Ismail conceived and designed the experiments, performed the experiments, prepared figures and/or tables, and approved the final draft.
- Shah Saud analyzed the data, authored or reviewed drafts of the article, and approved the final draft.
- Shah Hassan performed the experiments, analyzed the data, prepared figures and/or tables, and approved the final draft.
- Xuexuan Xu conceived and designed the experiments, performed the experiments, authored or reviewed drafts of the article, and approved the final draft.

## Data Deposition

The raw data is available in the Supplemental File.

## Supplemental Information

Supplemental information for this article can be found online at http://dx.doi.org/10.7717/peerj.16179#supplemental-information.

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
