# Peer review of "Enhancing wheat production and quality in alkaline soil: a study on the effectiveness of foliar and soil applied zinc"

_PeerJ, doi:10.7717/peerj.16179_

## Round 0.1 · original submission · Major Revisions

I have received comments from three independent reviewers and all of them stated that the article has scientific merit but needs substantial revision. Please revise the manuscript based on the comments of the reviewer along with a point-wise reply.

Reviewer 1 ·

Basic reporting

Comments and Suggestions
The manuscript presents a thorough and well-written examination of the topic. The issue of zinc deficiency is crucial to address as it has a significant impact on plant growth and development, leading to decreased crop yields. The manuscript provides valuable insights and information; however, there are some modifications that are needed to improve the manuscript.
Title:
The title “Comparative efficiency of foliar and soil applied zinc for optimum quality production of wheat in alkaline soil” is much simple and common. I suggest you to modify the title. If you decide to keep the same title, I would recommend you to change the part “optimum quality production”.
Abstract:
L-22. Zn was applied as ZnSO4 (Soil and foliar) at the time of sowing and tillering respectively, Please modify this line as I suggest “Zn was applied as zinc sulfate (ZnSO4) both in soil and as foliar spray at the time of sowing and tillering, respectively”.
L 27-28, therefore, it is very crucial for improving crop quality and yield and must be supplemented. Rewrite this sentence with correct grammar.
Introduction:
Write clear objectives of the experiment at the end of the introduction section. Also write the hypothesis of this research work.
Methodology:

CONCLUSIONS:
Modify the conclusions with correct grammar.

Experimental design

L 71-72, there were eleven (11) treatments. Remove the eleven from this line and write only 11.
Results:

Validity of the findings

Please correct the grammar of overall section of the results section.
L 100-101, corrects the unit of this number (3639 kg ha-1) and removes the word “application”.
L 102-103, the yield was higher in plots that received Zn as a soil with foliar spray. Modify this sentence with clear meaning.
L104-106, write “where” instead of “which” in these sentences and modify this sentences.
L146-147, Pearson correlation coefficients for activity of grain weight, tiller per plant, grain yield, biological yield, protein content, K in grain, P in grain, and Zn in grain content. Only write Pearsons’s correlation coefficients as a title of the table 4. Add the caption of the table 4

Reviewer 2 ·

Basic reporting

-The manuscript deals with Zn application for improving/optimizing its contents in wheat. It is a routine work, reported many times in the literature. What is the novelty of present work? So much has been done on this topic. Zn fortifying varieties have been developed. How this work adds values to the existing body of knowledge is missing?
-Authors have not taken into consideration recent literature. Nothing has been cited from 2021-2022. Introduction part is poorly written, it should be completed revised and updated.

Experimental design

-As mentioned RCBD was used for the experimentation but the authors have not explained how they achieved, randomization/blocking? A schematic can be added to the supplementary material.
-The soil selected and the title indicate that the soil was alkaline. With the pH and EC values of 7.6 and 0.32 dSm-1, was it really true?

Validity of the findings

The yield data indicate the production/grain yield well above the normal soil production average within Pakistan. The authors must provide supporting information to justify the findings.

Additional comments

-The abstract contains so many numbers that is not appropriate. It is better to provide ranges if possible.
-All the values in percentages should carry the symbol of percentage.
-Botanical/Technical names should be italicized.
-Based upon the outcomes of the present study, what are possible future perspectives? Please mention in conclusion part.

Reviewer 3 ·

Basic reporting

.

Experimental design

.

Validity of the findings

.

Additional comments

Title: Title is OK, however, only grain protein content was measured for wheat quality purpose.

Summary: Improvement required in text in terms of technical requirements and clarity of results. No information provided on alkaline soil and pH.
Line 30-32: Contradictory statements. All applications of Zn increased wheat yield, but a combined soil and foliar application of Zn showed pronounced effects.
Line 26-28: It is difficult to follow a large number of treatments. Better assign numbers like T1 and T10.
Line 29: ZnSO4. How many water molecules are in this formula?
Line 29: “Zn was applied as ZnSO4 (Soil and foliar) at the time of sowing and tillering respectively”. How the fertilizer doses were divided for each treatment at sowing and tillering?
Line 34: “Zn, N, P and K”. Expand all before using abbreviations.

Introduction: Text is poor and not organized. No information was given on the role of alkalinity on wheat growth and development. No citation on the effect of salinity on wheat. Rewrite introduction with suitable references.
Line 38: “Graminae” use current name Poaceae.
Line 39: Delete space between number and percent sign ‘36 %’.
Line 39: use ‘the’ before most.
Line 43: delete a. ‘and a lack’
Line 42: “Among abiotic factors, one of the primary causes of low yield is nutritional imbalance and a lack of essential macro and micro nutrients:. Citation required for this statement.
Line 45-46: Citation required for this statement.
Line 48: No sense. Rewrite this sentence. Zinc deficiency is an established 48 problem in field crops, because of decreased yield and nutritional quality.
Line 52: Do not start any sentence with brackets. ‘(Yoshida & Tanaka 1969) first recognized’
Line 53 and 56: Same as above. Fix it.
Line 52-56: Corn example was given for the effects of Zn. Delete it provide references on wheat such as:
Exogenous Application of Zinc Sulphate at Heading Stage of Wheat Improves the Yield and Grain Zinc Biofortification. Agronomy 2022, 12(3), 734; https://doi.org/10.3390/agronomy12030734. Published by Scientists from Pakistan
Line 38-69: Why no information or citation was provided on effects of alkaline soils on wheat growth and development.

Materials and Methods: Critical information is required in the methodology section.
Line 74: Clarify crop was grown for 1 year or 2 year?
Line 78-82: How many replicates of plants were used for observation?
What was the size of 33 or 66 plots?
How much Zn applied during sowing and at tillering? As mentioned in the abstract (Zn was applied as ZnSO4 (Soil and foliar) at the time of sowing and tillering respectively.).
What was the length of each row harvested?
How many replicates were used for N and Protein estimation?
When was the Foliar application of Zn applied?

Results: It is difficult to understand the results because it’s not clear when foliar application of Zn was conducted.
Line 105-106: Make 2 simple sentences. Difficult to understand the meaning of this sentence.
Line 107: unit (3639 kg ha-1). -1 must be in superscript form.
Line 119-120: So now authors used T1 and T10 terminology. Better use it from the very beginning and mention it in the abstract and method section.
Line 128-129: Do not use They for nutrients.

Discussion:
Line 158: Do not use fancy words. “greatly boosted”.
Not a single sentence in the entire discussion section showed how alkalinity affects any of the evaluated parameters.
Main concept of alkalinity is totally ignored. Authors need to rewrite discussion and keep focus around alkalinity instead of just discussing high and low concentration of Zn only.

Decision:
I recommend revision and resubmission.

---

## Round 0.2 · Major Revisions

First, the English language needs to be improved. Some sentences are incomprehensible.

There is no rationale for the study of zinc application to the soil and crop studied. The authors provide a figure of available zinc concentration, which they do not compare with the optimal or deficiency values. There are no data on concentrations of other available macro or micronutrients; that is, it is not clear that zinc is the growth-limiting nutrient and apparently no other nutrients were applied. Furthermore, the authors cite some published papers that pursue the same objectives in the same type of soil. The novelty of this manuscript is not clear. Neither in the discussion or conclusions are novel contributions with respect to those cited articles included. Hence, the Introduction needs substantial improvement.

The section “Materials and Methods” is incomplete and somewhat confusing. It should also be improved.

The Discussion includes some data that are in contradiction with what appears in the Tables. Some claims are made that are not explained. Some statements, which try to explain some data, are incorporated without citing the source. The Discussion must be substantially improved.

The Conclusions are, in some cases, not supported by the results.
I include detailed comments in the annotated manuscript.

Overall, the manuscript must be thoroughly revised.

Reviewer 2 ·

Basic reporting

No comments

Experimental design

No comment

Validity of the findings

No comment

Additional comments

The authors have revised the manuscript, addressed most of the comments and the quality of the manuscript has been improved significantly. There are some formatting errors that can be removed at proof stage.

---

## Round 0.3 · Major Revisions

The manuscript has somewhat been improved compared to the previous version. However, many of my recommendations have not been addressed and the manuscript still needs to be improved. Particularly, it is important to know if the available zinc concentration in the soil (the article presents this data) is far from or close to the sufficiency value. The authors say in their response that soil parameters were not their objective. However, it does not make sense to apply fertilizers without knowing the nutritional status of the soil. It should be known if not only the zinc but also the other nutrients concentrations are below the sufficiency level, since apparently no other fertilizers were applied.
The Material and Methods section is now cumbersome, but often unclear.
I provide a new annotated manuscript with detailed comments.

---

## Round 0.4 · Minor Revisions

The manuscript has been improved. I recommend only very minor revisions. Please see the annotated manuscript.

---

## Round 0.5 · accepted · Accept

The necessary modifications have been made. The manuscript is acceptable for publication.